# CONFIDE: CONTEXTUAL FINITE DIFFERENCE MODELLING OF PDES

## ABSTRACT

We introduce a method for inferring an explicit PDE from a data sample generated by previously unseen dynamics, based on a learned context. The training phase integrates knowledge of the form of the equation with a differential scheme, while the inference phase yields a PDE that fits the data sample and enables both signal prediction and data explanation. We include results of extensive experimentation, comparing our method to SOTA approaches, together with ablation studies that examine different flavors of our solution in terms of prediction error and explainability.

## 1 INTRODUCTION

Many scientific fields use the language of Partial Differential Equations (PDEs; Evans, 2010) to describe the physical laws governing observed natural phenomena with spatio-temporal dynamics. Typically, a PDE system is derived from first principles and a mechanistic understanding of the problem after experimentation and data collection by domain experts of the field. Well-known examples for such systems include Navier-Stokes and Burgers' equations in fluid dynamics, Maxwell's equations for electromagnetic theory, and Schrödinger's equations for quantum mechanics. Solving a PDE model could provide users with crucial information on how a signal evolves over time and space, and could be used for both prediction and control tasks.

Creating PDE-based models holds great value, but it is a difficult task in many cases. For some complex real-world phenomena, only some part of the systems dynamics is known, such as its structure, or functional form. For example, an expert might tell us that a signal obeys the dynamics of a heat equation, without specifying the diffusion and drift coefficient functions. We focus mainly on this case, as explained in detail below.

The current process of solving PDEs over space and time is by using numerical differentiation and integration schemes. However, numerical methods may require significant computational resources, making the PDE solving task feasible only for low-complexity problems, e.g., a small number of equations. An alternative common approach is finding simplified models that are based on certain assumptions and can roughly describe the problem's dynamics. A known example for such a model are the Reynolds-averaged Navier-Stokes equations Reynolds (1895). Building simplified models is considered a highly non-trivial task that requires special expertise, and might still not represent the phenomenon to a satisfactory accuracy.

In recent years, with the rise of Deep Learning (DL; LeCun et al., 2015), novel methods for solving numerically-challenging PDEs were devised. These methods have become especially useful thanks to the rapid development of sensors and computational power, enabling the collection of large amounts of multidimensional data related to a specific phenomenon. In general, DL based approaches consume the observed data and learn a black-box model of the given problem that can then be used to provide predictions for the dynamics. While this set of solutions has been shown to perform successfully on many tasks, it still suffers from two crucial drawbacks: (1) It offers no explainability as to why the predictions were made, and (2) it usually performs very poorly when extrapolating to unseen data.

In this paper, we offer a new hybrid modelling (Kurz et al., 2022) approach that can benefit from both worlds: it can use the vast amount of data collected on one hand, and utilize the partially known PDEs describing the observed natural phenomenon on the other hand. In addition, it can learn several

contexts, thus employing the generalization capabilities of DL models and enabling zero-shot learning (Palatucci et al., 2009).

Specifically, our model is given a general functional form of the PDE (i.e., which derivatives are used), consumes the observed data, and outputs the estimated coefficient functions. Then, we can then use off-the-shelf PDE solvers (e.g., PyPDE[1]) to solve and create predictions of the given task forward in time for any horizon.

Another key feature of our approach is that it consumes the spatio-temporal input signals required for training in an unsupervised manner, namely the coefficient functions that created the signals in the train set are unknown. This is achieved by combining an autoencoder architecture (AE; Kramer, 1991; Hinton & Salakhutdinov, 2006) with a loss defined using the functional form of the PDE. As a result, large amounts of training data for our algorithm can be easily acquired. Moreover, our ability to generalize to data corresponding to a PDE whose coefficients did not appear in the train set, enables the use of synthetic data for training. Although our approach is intended to work when the PDE functional form is known, it is not limited to that scenario only. In cases where we are given a misspecified model (when experts provide a surrogate model for instance), our model can eliminate some of the discrepancies using the extra function that is not a coefficient of one of the derivatives (the $p_0(x, t, u)$ function in equation 1).

On the technical side, we chose to apply a finite difference approach in order to integrate the knowledge regarding the structure of the PDE family. This approach enables us to consume training data without requiring the corresponding boundary conditions.

A natural question for this setup is whether we are able to extract the "correct" coefficients for the PDE. The answer depends on the identifiability of the system, a trait that does not hold for many practical scenarios. We therefore focus on finding the coefficients that best explain the data, making prediction of the signal forward in time possible. Practitioners will find the estimated coefficients useful even if they are not exact, since they may convey the shape, or dynamics, of unknown phenomena.

Our motivation comes from the world of electric vehicle batteries, where PDEs are used to model battery charging, discharging and aging. The data describing these phenomena is gathered by battery management systems in the vehicle, and also in the lab. Traditional techniques for model calibration suffer from two drawbacks: (1) they are extremely time consuming, (2) they do not leverage data from one battery in the dataset to another. Our approach solves both issues: model calibration is achieved by inference rather than optimization, and the learned context facilitates transfer of knowledge between batteries.

We summarize our contribution as follows:

1. Harnessing the information contained in large datasets belonging to a phenomenon which is related to a PDE functional family in an unsupervised manner. Specifically, we propose a regression based method, combined with a finite difference approach.

2. Proposing a DL encoding scheme for the context conveyed in such datasets, enabling generalization for prediction of unseen samples based on minimal input, similarly to zero-shot learning.

3. Extensive experimentation with the proposed scheme, examining the effect of context and train set size, along with a comparison to different previous methods.

The paper is organized as follows. In Section 2 we review related work. In Section 3 we present the proposed method and in Section 4 we provide experiments to support our method. Section 5 completes the paper with conclusions and future directions.

## 2    RELATED WORK

Creating a neural-network based model for approximating the solution of a PDE has been studied extensively over the years, and dates back more than two decades (Lagaris et al., 1998). We divide deep learning based approaches by their ability to incorporate mechanistic knowledge in their models, and by the type of information that can be extracted from using them. Another distinction between

---

[1] https://pypde.readthedocs.io/en/latest/

different approaches is their ability to handle datasets originating from different contexts. From a PDE perspective, a different context could refer to having data signals generated with different coefficients functions ($p_l$ in equation 1). In many real-world applications, obtaining observed datasets originating from a single context is impractical. For example, in cardiac electrophysiology (Neic et al., 2017), patients differ in cardiac parameters like resistance and capacitance, thus representing different contexts. In fluid dynamics, the topography of the underwater terrain (bathymetry) differs from one sample to another (Hajduk et al., 2020).

The first line of work is purely data-driven methods. These models come in handy when we observe a spatio-temporal phenomenon, but either don't have enough knowledge of the underlying PDE dynamics, or the known equations are too complicated to solve numerically (as explained thoroughly by Wang & Yu (2021)). Recent advances demonstrate successful prediction results that are fast to compute (compared to numerically solving a PDE), and also provide decent predictions even for PDEs with very high dimensions (Brandstetter et al., 2022; Li et al., 2020; Han et al., 2018; Lu et al., 2019; Pfaff et al., 2020). However, the downside of these approaches is not being able to infer the PDE coefficients, which may hold valuable information and explanations as to why the model formed its predictions.

The second type of data-driven methods are approaches that utilize PDE forms known beforehand to some extent. Works that adopt this approach can usually utilize the given mechanistic knowledge and provide reliable predictions, ability to generalize to unseen data, and in some cases even reveal part of the underlying PDE coefficient functions. However, their main limitation is that they assume the entire training dataset is generated by a single coefficient function and only differ in the initial conditions (or possibly boundary conditions). PDE-NET (Long et al., 2018), its followup PDE-NET2 (Long et al., 2019), DISCOVER (Du et al., 2022), PINO (Li et al., 2021) and sparse-optimization methods (Schaeffer, 2017; Rudy et al., 2017) (expanding the idea originally presented on ODEs Brunton et al. (2016); Champion et al. (2019)), are not given the PDE system, but instead aim to learn some representation of the underlying PDE as a linear combination of base functions and derivatives of the PDE state. PINN (Raissi et al., 2019) and NeuralPDE (Zubov et al., 2021) assume full knowledge of the underlying PDE including the its coefficients, and aim to replace the numerical PDE solver by a fast and reliable model. They also provide a scheme for finding the PDE parameters as scalars, but assume the entire dataset is generated by a single coefficient value, while we assume each sample is generated with different coefficient values which could be functions of time, space and state (as described in equation 1). In Négiar et al. (2022), the authors incorporate knowledge of the PDE structure as a hard constraint while learning to predict the solution to the PDE. Similarly, Learning-informed PDEs (Dong et al., 2022; Aarset et al., 2022) suggest a method that assumes full knowledge of the PDE derivatives and their coefficient functions, and infers the free coefficient function (namely $p_0(x, t, u)$ in equation 1). In (Lim et al., 2022), the authors apply a finite difference approach to PINNs. Another approach for learning the solution to PDEs, that also uses a neural representation, is introduced in (Chen et al., 2022). Recently, in (Subramanian et al., 2023), the authors have suggested a Foundation Model framework for predicting solutions of PDEs.

The last line of work, and closer in spirit to ours, includes context-aware methods that assume some mechanistic knowledge, with each sample in the train set generated by different PDE coefficients (we also refer to this concept as having different context) and initial conditions. CoDA (Kirchmeyer et al., 2022) provides the ability to form predictions of signals with unseen contexts, but does not directly identify the PDE parameters. GOKU (Linial et al., 2021) and ALPS (Yang et al., 2022) provide context-aware inference of signals with ODE dynamics, when the observed signals are not the ODE variables directly. Another important paper introduces the APHYNITY algorithm (Yin et al., 2021), which also presents an approach to inferring PDE parameters from data. This work handles the scenario of fixed coefficients, as opposed to our ability to handle coefficients that are functions. Also, the case of coefficients that differ between samples is addressed only briefly, with a fixed, rather high, context ratio.

## 3 METHOD

The data we handle is a set of spatio-temporal signals generated by an underlying PDE, only the form of which is known. The coefficient functions determining the exact PDE are unknown and may be different for each collection of data. Our goal is to estimate these coefficient functions and

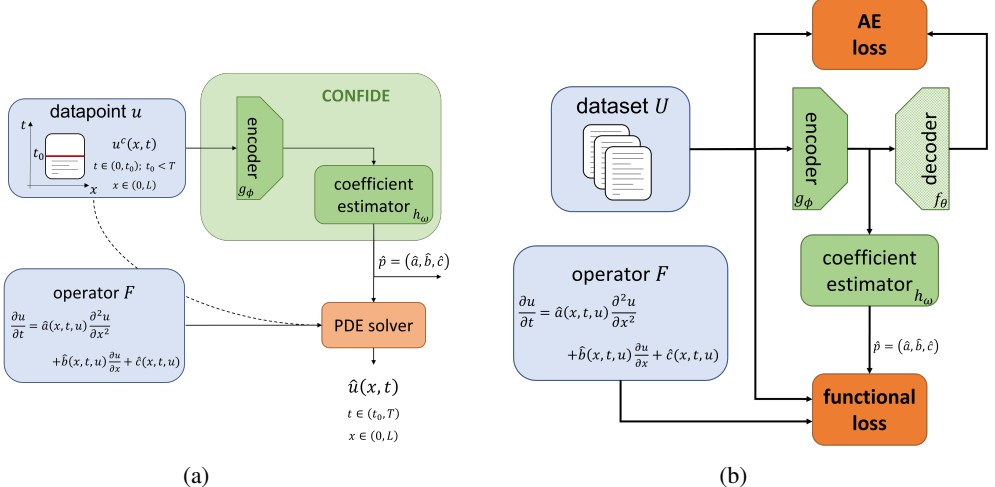

Figure 1: (a) Inference process. Dashed line is initial condition. (b) Training process.

provide reliable predictions of the future time steps of the observed phenomenon. The proposed method comprises three subsequent parts: (1) Creating a compact representation of the given signal, (2) estimating the PDE coefficients, and (3) solving the PDE using the acquired knowledge. For ease of exposition we focus on parabolic PDEs in this section.

### 3.1 PROBLEM FORMULATION

We now define the problem formally. Let $u(x, t)$ denote a signal with spatial support $x \in [0, L]$ and temporal support $t \in [0, T]$. We refer to this as the *complete* signal. Next, we define $u^c(x, t)$ to be a partial input signal, a *patch*, where its support is $x \in [0, L]$, $t \in [0, t_0]$, $0 < t_0 < T$. The superscript $c$ stands for context. We assume the signal $u(x, t)$ is the solution of a $k$-order PDE of the general form

$$\frac{\partial u}{\partial t} = \sum_{l=1}^{k} p_l(x, t, u) \frac{\partial u^l}{\partial x^l} + p_0(x, t, u),$$  (1)

with a vector of coefficient functions $p = (p_0, \ldots, p_k)$. We adopt the notation of Wang & Yu (2021) and refer to a family of PDEs characterized by a vector $p$ as an operator $F(p, u)$, where solving $F(p, u) = 0$ yields solutions of the PDE.

The problem we solve is as follows: given a patch $u^c(x, t)$, that solves a PDE of a *known* operator $F$ with an *unknown* coefficient vector $p$, we would like to (a) estimate the coefficient vector $\hat{p}$ and (b) predict the complete signal $\hat{u}(x, t)$ for $0 \leq t \leq T$.

Our solution is a concatenation of two neural networks, which we call *CONFIDE*. Its input is a patch, and its output is a vector $\hat{p}$. We feed this vector into an off-the-shelf PDE solver together with the operator $F(p, u)$ to obtain the predicted signal $\hat{u}(x, t)$. An explanation of our numerical scheme appears in Section A.

### 3.2 CONFIDE INFERENCE

We begin by outlining our inference process, presented in Fig. 1a. The input to this process is a patch $u^c(x, t)$, where $x \in [0, L]$, $t \in [0, t_0]$ and an operator $F$ (e.g., the one introduced in equation 1 for $k = 2$). The patch is fed into the CONFIDE component, which generates the estimated coefficients $\hat{p}$, in the example, $\hat{p} = (\hat{a}, \hat{b}, \hat{c})$. The PDE solver then uses this estimate to predict the complete signal, $\hat{u}(x, t)$, $x \in [0, L]$, $t \in [t_0, T]$. An important feature of our approach is the explicit prediction of the coefficient functions, which contributes to the explainability of the solution.

The patch $u^c(x, t)$ is a partial signal that serves as an initial condition for the prediction and also represents the dynamics of the signal for estimating the PDE coefficients. In the sequel we refer to it

as "context". The ratio of the context is denoted by $\rho$, such that $t_0 = \rho T$, and is a hyper-parameter of our algorithm. We discuss the effect of context size in Section D.2.

---

**Algorithm 1** CONFIDE inference scheme

---

**Input:** patch $u^c(x, t)$, operator $F$, trained networks: decoder $g_\phi$, coefficient estimator $h_\omega$
$\hat{p} \leftarrow h_\omega(g_\phi(u^c))$
$\hat{u} \leftarrow \text{PDE\_solve}(F, \hat{p}, u^c(x, t = t_0))$
return $\hat{u}, \hat{p}$

---

**Algorithm 2** Algorithm for training CONFIDE

---

**Input:** dataset $U$, operator $F$, context ratio $\rho$, loss weight $\alpha$, number of epochs $N_e$
**Init:** random weights in encoder $g_\phi$, decoder $f_\theta$, coefficient estimator $h_\omega$
**for** epoch in $N_e$ **do**
    $\mathcal{L} \leftarrow 0$
    $U_N^c \leftarrow N$ random patches, one from each $u_i \in U$
    **for** $u_i^c$ in $U_N^c$ **do**
        $\hat{p}_i \leftarrow h_\omega(g_\phi(u_i^c))$
        $\mathcal{L}_{\text{AE}} \leftarrow (u_i^c - f_\theta(g_\phi(u_i^c), u_i(t = 0)))^2$
        $\mathcal{L}_{\text{coef}} \leftarrow \|F(\hat{p}_i, u_i^c)\|^2$
        $\mathcal{L} \leftarrow \mathcal{L} + \alpha \cdot \mathcal{L}_{\text{AE}} + (1 - \alpha) \cdot \mathcal{L}_{\text{coef}}$
    **end for**
    $\phi, \theta, \omega \leftarrow \arg \min \mathcal{L}$
**end for**

---

### 3.3 CONFIDE TRAINING

The training process is presented in Fig. 1b. Its input is a dataset $U$ that consists of $N$ complete signals $\{u_i(x, t)\}_{i=1}^N$, which are solutions of $N$ PDEs that share an operator $F$ but have unique coefficient vectors $\{p_i\}_{i=1}^N$. We stress that the vectors $p_i$ are unknown even at train time. The support of the signals is $x \in [0, L]$, $t \in [0, T]$. The loss we minimize is a weighted sum of two components: (i) the autoencoder reconstruction loss , which is defined in equation 2, and (ii) the functional loss as defined in equation 3.

CONFIDE comprises two parts: (1) an encoder and (2) a coefficient estimator. The encoder's goal is to capture the dynamics driving the signal $u_i$, thus creating a compact representation for the coefficient estimator. The encoder is trained on patches $u_i^c$ randomly taken from signals $u_i$ belonging to the train set. Each patch is of size $t_0 \times L$.

The encoder loss is the standard AE reconstruction loss, namely the objective is

$$\min_{\theta, \phi} \mathcal{L}_{\text{AE}} = \min_{\theta, \phi} \sum_{i=1}^N \text{loss}(u_i^c - f_\theta(g_\phi(u_i^c))), \tag{2}$$

where $f_\theta$ is the decoder, $g_\phi$ is the encoder and $\text{loss}(\cdot, \cdot)$ is a standard loss function (e.g., $L^2$ loss).

The second component is the coefficient estimator, whose input is the encoded context. The estimated coefficients output by this component, together with the operator $F$, form the functional objective:

$$\min_\omega \mathcal{L}_{\text{coef}} = \min_\omega \sum_{i=1}^N \|F(\hat{p}_\omega, u_i^c)\|^2, \tag{3}$$

where $\omega$ represents the parameters of the coefficient estimator network, and $\hat{p}$ is the estimator of $p$, acquired by applying the network $h_\omega$ to the output of the encoder.

The two components are trained simultaneously, and the total loss is a weighted sum of the losses in equation 2 and equation 3: $\mathcal{L} = \alpha \cdot \mathcal{L}_{\text{AE}} + (1 - \alpha) \cdot \mathcal{L}_{\text{coef}}$, where $\alpha \in (0, 1)$ is a hyper-parameter.

**Initial-conditions aware autoencoder**. To further aid our model in learning the underlying dynamics of the observed phenomenon, we include the observed initial conditions of the signal (i.e., $u_i(t = 0)$)

along with the latent context vector (i.e., $g_\phi(u_i^c)$) as input to the decoder network. This modification enables the model to learn a context vector that better represents the dynamics of the phenomenon, rather than other information such as the actual values of the signal.

We experimented with removing the decoder and training the networks using the functional loss alone, and without including the initial conditions as an input to the decoder. In both cases, results proved to be inferior, suggesting that the autoencoder loss helps the model to focus on the underlying dynamics of the observed signal.

To summarize this section, we present the inference scheme in Algorithm 1, and the full training algorithm in Algorithm 2.

## 4 EXPERIMENTS

We devote this section to analyse and compare our approach to other solutions, on three different systems of PDEs: (1) constant coefficients, (2) Burgers' equations, and (3) 2D-FitzHugh-Nagumo. For each PDE task, we created a dataset of signals generated from a PDE with different coefficients. We could not use off-the-shelf datasets, such as those appearing in PDEBench (Takamoto et al., 2022), since each of the datasets there is generated from a single constant function (i.e., all data samples have the same context). We used well-known equations, therefore our datasets can serve as a benchmark for the emerging field of contextual PDE modelling. We stress the fact that the test set contains signals generated by PDEs with coefficient vectors that *do not* appear in the training data, resulting in a zero-shot prediction problem. More information about dataset creation can be found in the appendix.

We benchmark the performance of CONFIDE against several state of the art approaches:

1. Neural ODE, based on the algorithm suggested by Chen et al. (2018), Section 5.1 (namely, Latent ODE).
2. Fourier Neural Operator (FNO), introduced by Li et al. (2020).
3. U-Net, as presented by Gupta & Brandstetter (2022).
4. DINO, as presented by Yin et al. (2022).

Additional details regarding the implementation of baselines can be found in Section B.2.

### 4.1 SECOND ORDER PDE WITH CONSTANT COEFFICIENTS

The first family of PDEs used for our experiments is:

$$\frac{\partial u}{\partial t} = a\frac{\partial^2 u}{\partial x^2} + b\frac{\partial u}{\partial x} + c, \tag{4}$$

where $p = (a, b, c)$ are constants but differ between signals. Figure 2a demonstrates the clear advantage of our approach, which increases with the prediction horizon (note the logarithmic scale of the vertical axis, representing the MSE of prediction). Since CONFIDE harnesses both mechanistic knowledge and training data, it is able to predict the signal $\hat{u}(x, t)$ several timesteps ahead, while keeping the error to a minimum.

Another result for this set of experiments appears in Figure 2b. Here, we plot the estimated value of parameter $a$ of equation 4, against its true value. The plot and the high value of $R^2$ demonstrate the low variance of our prediction, with a strong concentration of values along the $y = x$ line.

Section D presents the results of an ablation study on the hyper-parameters of CONFIDE for this equation.

### 4.2 BURGERS' EQUATION

Another family of PDEs we experiment with is the quasi-linear Burgers' equation, whose general form is

$$\frac{\partial u}{\partial t} = a\frac{\partial^2 u}{\partial x^2} + b(u)\frac{\partial u}{\partial x}, \tag{5}$$

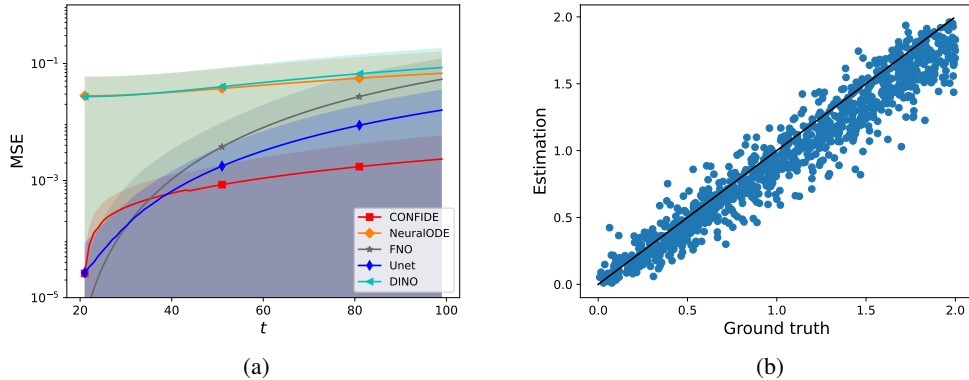

(a)                                              (b)

Figure 2: Constant coefficients (Section 4.1). **(a)** Prediction error vs. prediction horizon, for different algorithms. CONFIDE, in red, is our approach. **(b)** Estimated value of the $\partial^2 u / \partial x^2$ coefficient vs. ground truth, for test set ($R^2 = 0.93$).

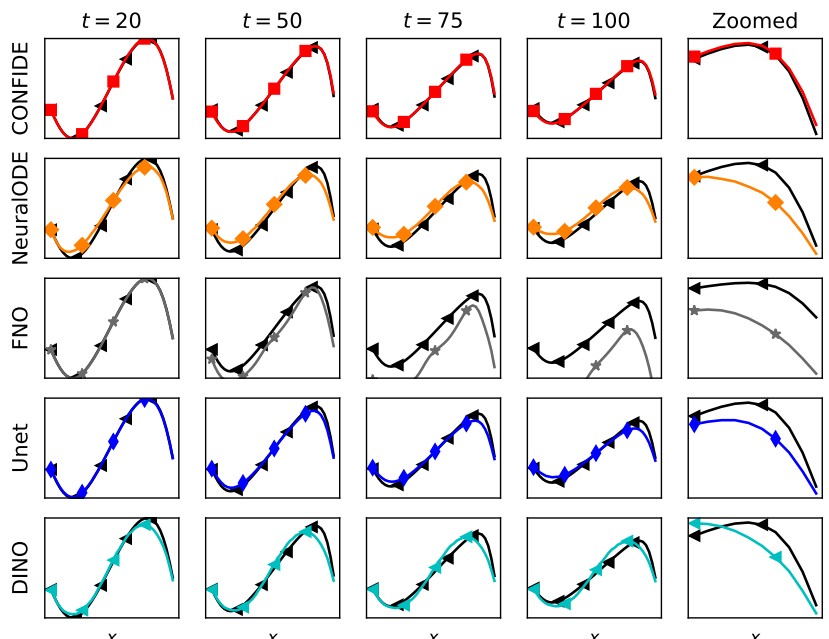

Figure 3: A solution of the Burgers' equation. The black plot in each figure displays the ground truth. Rows correspond with the predicted solution by the respective algorithm (top row for CONFIDE displayed in red). Each column shows the solution at a different time point. The rightmost column shows the solution at $t = 100$ zoomed to demonstrate the differences.

where $b(x, t, u) = -u$, as presented in Bateman (1915). We note that this equation is quasi-linear since its drift coefficient $b(x, t, u)$ depends on the solution $u$ itself. The dataset for our experiments consists of 10000 signals with different values of $a$ and the same $b(u) = -u$, both unknown to the algorithm a priori. We begin with a demonstration of a signal $u(x, t)$ and its prediction $\hat{u}(x, t)$ in Figure 3. As can be seen both visually and from the value of the MSE (in each panel's title), our approach yields a prediction that stays closest to the ground truth (GT), even as the prediction horizon (vertical axis) increases.

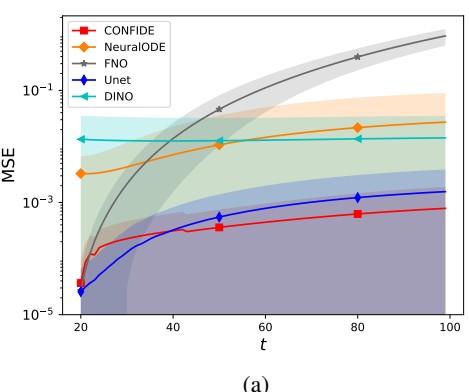 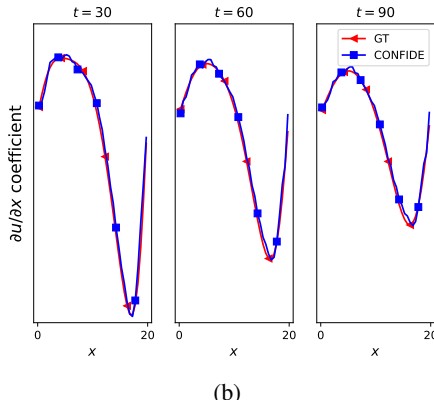

(a)                                         (b)

Figure 4: Burgers' PDE: **(a)** Prediction error as prediction horizon increases, for different approaches with context ratio $\rho = 0.2$. **(b)** Estimation of the coefficient function $b(x, t, u)$ of the Burgers' equation from equation 5. CONFIDE manages to accurately estimate the spatio-temporal dynamics of the coefficient, based on a context ratio of $\rho = 0.2$.

Figure 4a displays a comparison between the different approaches to our problem. As before, the vertical axis of the plot is logarithmic, and the advantage of CONFIDE over other approaches increases with the prediction horizon. In Figure 4b we focus on the ability to accurately predict coefficient functions with spatio-temporal dynamics, in this case: the coefficient $b(x, t, u)$ of equation 5. The panels correspond to different points in time, showing that the coefficient estimator tracks the temporal evolution successfully.

### 4.3 FITZHUGH-NAGUMO EQUATIONS

The last family of PDEs we examine is the FitzHugh-Nagumo PDE (Klaasen & Troy, 1984) consisting of two equations:

$$\frac{\partial u}{\partial t} = a\Delta u + R_u(u, k, v), \qquad \frac{\partial v}{\partial t} = b\Delta v + R_v(u, v), \tag{6}$$

where $a$ and $b$ represent the diffusion coefficients of $u$ and $v$, and $\Delta$ is the Laplace operator. For the local reaction terms, we follow Yin et al. (2021) and set $R_u(u, k, v) = u - u^3 - k - v$, and $R_v(u, v) = u - v$. The PDE state is $(u, v)$, defined on the 2-D rectangular domain $(x, y)$ with periodic Neumann boundary conditions.

The dataset created for this task consists of 1000 signals, each with a different value of $k$. We compare the prediction generated by CONFIDE to those yielded by other approaches, and present a typical result in Fig. 5. In Figure 7 we present the prediction error as a function of the prediction horizon, once again comparing CONFIDE to the baselines.

We summarize the results of experiments for signal prediction across all setups and approaches in Table 2. The table includes results for CONFIDE, all baselines, and also a variant of CONFIDE which we refer to as CONFIDE-0. This zero-knowledge variant is applicable when we know that the signal obeys some differential operator $F$, but have no details regarding the actual structure of $F$. Thus, CONFIDE-0 does not estimate the equation parameters, and only yields a prediction for the signal, utilizing our context-based architecture. We elaborate further in Section B.2.

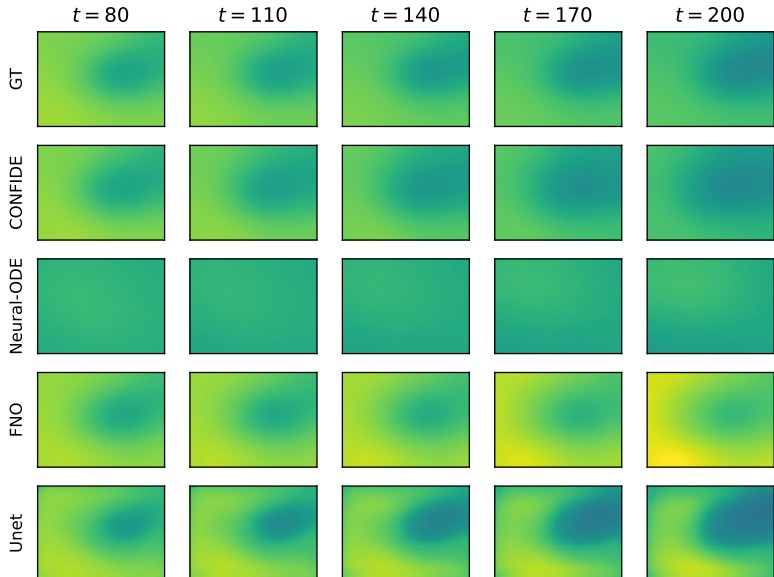

Figure 5: Figures in the top row show the ground truth of $R_v$ for different time points, and the rows below show the estimation of it by the different approaches. CONFIDE near-perfectly recovers the unknown part of the PDE even as the prediction horizon increases.

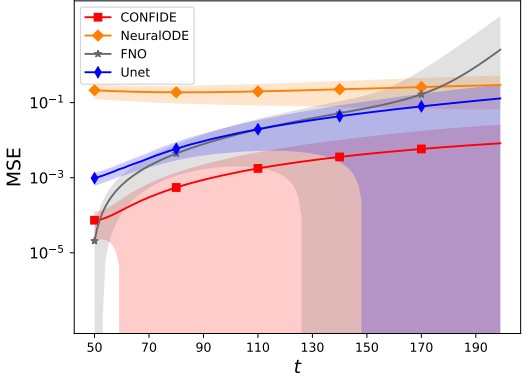

Figure 6: 2D-FitzHugh-Nagumo PDE: prediction error as horizon increases, for different approaches.

| Setup | Coefficient estimation error |
|---|---|
| Constant coeff. | $0.0095 \pm 0.0131$ |
| Burgers' | $0.0454 \pm 0.0333$ |
| FN2D | $0.0075 \pm 0.0123$ |

Table 1: Coefficient estimation error for different experimental setups: constant coefficients, Burgers' equation and two-dimensional FitzHugh-Nagumo. The variance is calculated over the entire test set, namely 1000 signals for the first two setups and 100 signals for FN.

## 5   CONCLUSION

In this work we introduce a new hybrid modelling approach, combining mechanistic knowledge with data. The knowledge we assume is in the form of a PDE family, without specific parameter values, typically supplied by field experts. The dataset we rely on is readily available in physical modelling

Table 2: Result summary for the signal prediction task, on all three PDE systems. The numbers represent signal prediction error at the end of the prediction horizon, averaged over the entire test set.

| Method | Constant coefficients | Burgers' | FitzHugh-Nagumo |
|---|---|---|---|
| CONFIDE | $0.0023 \pm 0.0036$ | $0.0008 \pm 0.0011$ | $0.0083 \pm 0.0177$ |
| CONFIDE-0 | $0.0079 \pm 0.0218$ | $0.0009 \pm 0.0016$ | $0.0845 \pm 0.0978$ |
| Neural-ODE | $0.0680 \pm 0.0905$ | $0.0272 \pm 0.0627$ | $0.2944 \pm 0.2293$ |
| FNO | $0.0538 \pm 0.0680$ | $0.9351 \pm 0.3091$ | $2.5727 \pm 17.732$ |
| Unet | $0.0160 \pm 0.0199$ | $0.0016 \pm 0.0023$ | $0.1293 \pm 0.1748$ |
| DINO | $0.0850 \pm 0.0994$ | $0.0142 \pm 0.0206$ | N/A |

problems, as it is simply a collection of spatio-temporal signals belonging to the same PDE family, with different parameters. Unlike other schemes, we do not require knowledge of the parameters of the PDE generating our train data. We conduct extensive experiments, comparing our scheme to other solutions and testing its performance in different regimes. It achieves good results in the zero-shot learning problem, and is robust to different values of hyper-parameters.

Future directions we would like to pursue include a straightforward extension to handle signals with missing datapoints, handling "out of distribution" signals, generated by parameters beyond the support of the dataset, and examining the robustness of predicting such signals. Another question that comes to mind is whether including multiple signals generated by the same parameters has an effect on quality of results, similar to or different from that of the context ratio. Finally, we are eager to apply CONFIDE to a real world problem like the ones mentioned in Section 2.

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

## A  NUMERICAL SCHEME

The partial derivatives are estimated using standard numerical schemes for each point in the patch. We choose discretization parameters $\Delta x$ for the spatial axis and $\Delta t$ for the temporal axis where we solve the PDE numerically on the grid points $\{(i\Delta x, j\Delta t)\}_{i=0,j=0}^{N_x,N_t}$ with $L = N_x\Delta x$ and $T = N_t\Delta t$. Let us denote the numerical solution with $\hat{u}_{i,j}$. We use the *forward-time central-space* scheme, so a second order scheme from equation 1 would be

$$
\begin{aligned}
\frac{\hat{u}_{i,j+1} - \hat{u}_{i,j}}{\Delta t} =& p_2(i,j,u(i,j))\frac{\hat{u}_{i+1,j} - 2\hat{u}_{i,j} + \hat{u}_{i-1,j}}{\Delta x^2} \\
& + p_1(i,j,u(i,j))\frac{\hat{u}_{i+1,j} - \hat{u}_{i-1,j}}{2\Delta x} \\
& + p_0(i,j,u(i,j))
\end{aligned}
\tag{7}
$$

We refer the reader to Strikwerda (2004) for a complete explanation.

## B  EXPERIMENTAL AND IMPLEMENTATION DETAILS

We provide further information regarding the experiments described in Section 4. We ran all of the experiments on a single GPU (NVIDIA GeForce RTX 2080), and all training algorithms took $< 10$ minutes to train. All algorithms used 5-10M parameters (more parameters on the FitzHugh-Nagumo experiment). Full code implementation for creating the datasets and implementing CONFIDE and its baselines will be made available upon acceptance.

### B.1  DATASET DETAILS

To create the dataset, we generated signals using the `PyPDE` package, where each signal was generated with different initial conditions. In addition, as discussed in Section 2, we made an important change that makes our setting much more realistic than the one used by other known methods: the PDE parametric functions (e.g., $(a, b, c)$) are sampled for each signal, instead of being fixed across the dataset, making the task much harder. To evaluate different models on the different datasets, we divided the datasets into 80% train set, 10% validation set and 10% test set.

**Second Order PDE with Constant Coefficients.** For this task, we generated 10,000 signals on the spatial grid $x \in [0, 20]$ with $\Delta x = 0.5$, resulting in a spatial dimension consisting of 40 points. Each signal was generated with different initial conditions sampled from a Gaussian process posterior that obeys the Dirichlet boundary conditions $u(x = 0) = u(x = L) = 0$. The hyper-parameters we used for the GP were $l = 3.0, \sigma = 0.5$, which yielded a rich family of signals, as demonstrated in Fig. 7a. The parameter vector was sampled uniformly: $a \sim U[0, 2]$, $b$ and $c \sim U[-1, 1]$ for each signal, resulting in various dynamical systems in a single dataset. To create the signal we solved the PDE numerically, using the explicit method for times $t \in [0, 5.0]$ and $\Delta t = 0.05$. Signals that were numerically unstable were omitted and regenerated, so that the resulting dataset contains only signals that are physically feasible.

**Burgers' PDE.** To create the Burgers' PDE dataset we followed the exact same process as with the constant coefficients PDE, except for the parameter sampling method. Parameter $a$ was still drawn uniformly: $a \sim U[1, 2]$, but $b$ here behaves as a function of $u$: $b(u) = -u$, commonly referred to as the viscous Burgers' equation.

**FitzHugh-Nagumo equations.** For the purpose of creating a more challenging dataset with two spatial dimensions we followed Yin et al. (2021), and used the 2-D FitzHugh-Nagumo PDE (described in Eq. 6). To make this task even more challenging and realistic, we created a small dataset comprising only 1000 signals defined on a 2D rectangular domain, discretized to the grid $[-0.16, 0.16] \times [-0.16, 0.16]$. The initial conditions for each signal were generated similarly to the other experiments, by sampling a Gaussian process prior with $l = 0.1$, which generated a rich family of initial conditions, as can be seen in Fig. 7b. To create the coefficient function we sample $k \sim U[0, 1]$ per signal, and set $(a, b) = (1e-3, 5e-3)$. To create the signal we solved the PDE numerically, using the explicit method for times $t \in [0, 1.0]$ and $\Delta t = 0.01$.

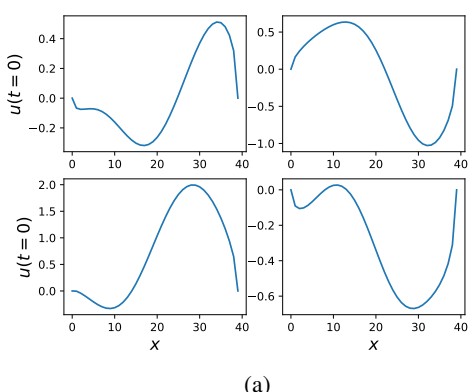 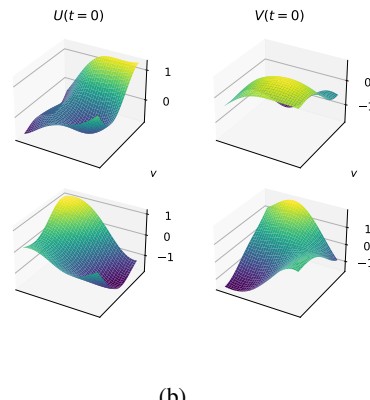

(a)                (b)

Figure 7: Demonstration of the rich family of initial conditions generated. **(a)** Four examples for different initial conditions generated for the Second order PDE with constant coefficients dataset. The Dirichlet boundary conditions are $u(x = 0) = u(x = L) = 0$, and the initial conditions are drawn from a GP posterior obeying the boundary conditions. **(b)** Two examples of initial conditions for the 2D FitzHugh-Nagumo datasets. The left column describes the first state variable $u$, and the right column is the state variable $v$. Top row is the first example, and the bottom row is the second. All initial conditions are drawn from a GP prior not constrained to boundary conditions.

### B.2 IMPLEMENTATION DETAILS

**CONFIDE.** The CONFIDE algorithm consists of two main parts: an auto-encoder part that is used for extracting the context, and a coefficient-estimation network.

The autoencoder architecture consists of an encoder-decoder network, both implemented as MLPs with 6 layers and 256 neurons in each layer, and a ReLU activation. For the FitzHugh-Nagumo dataset, we wrap the MLP autoencoder with convolution and deconvolution layers for the encoder and decoder respectively, in order to decrease the dimensions of the observed signal more effectively. We note that the encoder-decoder architecture itself is not the focus of the paper. We found that making the autoencoder initial-conditions-aware by concatenating the latent vector in the output of the encoder to the initial conditions of the signal $u(t = 0)$, greatly improved results and convergence time. The reason is that it encourages the encoder to focus on the dynamics of the observed signal, rather than the initial conditions of it. We demonstrate this effect in Section D.

The second part, which is the coefficient estimator part, is implemented as an MLP with 5 hidden layers, each with 1024 neurons, and a ReLU activation. The output of this coefficient-estimator network is set to be the parameters for the specific task that is being solved. In the constant-parameters PDE, the output is a 3-dim vector $(\hat{a}, \hat{b}, \hat{c})$. In the Burgers' PDE, the output is composed of a scalar $\hat{a}$, which is the coefficient of $\frac{\partial^2 u}{\partial x^2}$ and the coefficient function $b(u)$, which is a vector approximating the coefficient of $\frac{\partial u}{\partial x}$ on the given grid of $x$. In the FitzHugh-Nagumo PDE, the output is a scalar $k$ used for inferring $R_u(u, v, k)$, and the function $R_v$ on the 2D grid $(x, y)$.

The next step in the CONFIDE algorithm is to evaluate the loss which is comprised of two losses: an autoencoder reconstruction loss $\mathcal{L}_{AE}$, and a PDE functional loss $\mathcal{L}_{coef}$. The autoencoder loss is a straightforward $L^2$ evaluation on the observed signal $u^c$ and the reconstructed signal. The functional loss is evaluated by first numerically computing all the derivatives of the given equation on the observed signal. Second, evaluating both sides of the differential equations using the derivatives and the model's coefficient outputs, and lastly, minimizing the difference between the sides. For example, in the Burgers' equation, we first evaluate $\frac{\partial u}{\partial t}$, $\frac{\partial^2 u}{\partial x^2}$, and $\frac{\partial u}{\partial x}$, we then compute the coefficients $\hat{a}$ and $\hat{b}(u)$, and finally minimize:

$$\min_{\omega} \left\| \frac{\partial u}{\partial t} - \hat{a} \cdot \frac{\partial^2 u}{\partial x^2} - \hat{b}(u) \cdot \frac{\partial u}{\partial x} \right\|.$$

Since this algorithm evaluates numerical derivatives of the observed signals, it could be used for equations with higher derivatives, such as the wave equation, for instance.

**CONFIDE-0.** Similarly to the standard CONFIDE algorithm, we consider a zero-knowledge version, where we only know that the signal obeys some differential operator $F$, but have no details regarding the actual structure of $F$. Thus, the input for the coefficient-estimator network is the current PDE state ($u$ in the 1D experiment and $(u, v)$ in the 2D experiment), and the latent vector extracted from the auto-encoder. The model then outputs an approximation for time derivative of the PDE states, i.e., the model's inputs are $(u_t, g_\phi(u^c))$ and the output is an approximation for $\frac{\partial u}{\partial t}$. The optimization function for this algorithm therefore tries to minimize the difference between the numerically computed time derivative and the output of the model:

$$\mathcal{L}_{\text{CONFIDE-0}} = \alpha \cdot \mathcal{L}_{AE} + (1 - \alpha) \cdot \sum_{i=1}^{N} \left\| \frac{\partial u}{\partial t} - m_\theta(u_i^c, g_\phi(u_i^c)) \right\|^2,$$

where $\mathcal{L}_{AE}$ is defined in Eq. 2, $\frac{\partial u}{\partial t}$ is evaluated numerically, $m_\theta$ is the network estimating the temporal derivative, $g_\phi$ is the encoder network, and $u_i^c$ is the observed patch.

**Hyper-parameters** For both versions of CONFIDE we used the standard Adam optimizer, with learning rate of $1e^{-3}$, and no weight decay. For all the networks we used only linear and convolution layers, and only used the ReLU activation functions. For the $\alpha$ parameter we used $\alpha = 0.5$ for all experiments, and all algorithms, after testing only two different values: 0 and 0.5 and observing that using the autoencoder loss helps scoring better and faster results.

**Neural-ODE** We implement the Neural-ODE algorithm as suggested by Chen et al. (2018), section 5.1 (namely, Latent-ODE). We first transform the observed signal through a recognition network which is a 6-layer MLP. We then pass the signal through an RNN network backwards in time. The output of the RNN is then divided into a mean function, and an std function, which are used to sample a latent vector. The latent vector is used as initial conditions to an underlying ODE in latent space which is parameterized by a 3-layer MLP with 200 hidden units, and solved with a DOPRI-5 ODE-solver. The output signal is then transformed through a 5-layer MLP with 1024 hidden units, and generates the result signal. The loss function is built of two terms, a reconstruction term and a KL divergence term, which is multiplied by a $\lambda_{KL}$. After testing several optimization schemes, including setting $\lambda_{KL}$ to the constant values $\{1, 0.1, 0.01, 0.001, 0\}$, and testing a KL-annealing scheme where $\lambda_{KL}$ changes over time, we chose $\lambda_{KL} = 1e^{-2}$ as it produced the lowest reconstruction score on the validation set. We used an Adam optimizer with $1e^{-3}$ learning rate and no weight decay.

Our implementation is based on the code in `https://github.com/rtqichen/torchdiffeq`.

**FNO and Unet** For the Fourier-Neural-Operator we used the standard Neural-Operator package in `https://github.com/neuraloperator/neuraloperator`. For the Unet implementation we used the implementation in `https://github.com/microsoft/pdearena`. The input we used for both of these algorithms is the entire context $u_c$ from time $t = 0$ to $t = T - 2$, and the output is a prediction of the solution at the next time point $u(t = T - 1)$. The loss is therefore an MSE reconstruction loss on $u(t = T - 1)$.

## C  OUT-OF-DISTRIBUTION DATA

In this section, we provide additional experiments conducted on out-of-distribution (OOD) data. These experiments where selected to demonstrate how CONFIDE can handle observations that are significantly different than the data in the train set. We divide the OOD experiments into two parts: (1) the initial conditions observed are not smooth and have some discontinuity, and (2) the parameters used to generate the signals in the test set are sampled from a different distribution than the one used in the train set.

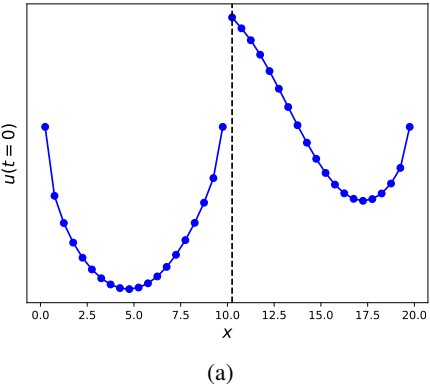
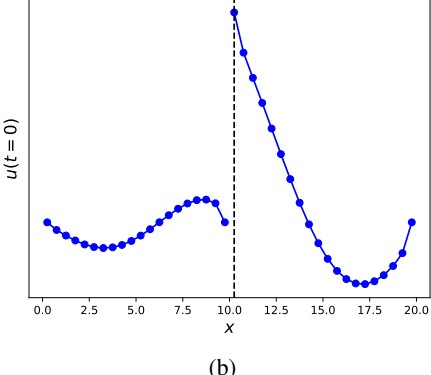

(a)                                         (b)

Figure 8: Demonstration of two initial conditions of shock-wave signals from the test set. Both signals have a discontinuity in $x = L/2$ (for $L = 20$).

### C.1  NON-SMOOTH INITIAL CONDITIONS

In this first benchmark, we demonstrate how CONFIDE handles the case where the observed data has a non smooth point, which resembles a shock-wave in the observations. Testing CONFIDE on non-smooth signals is important mainly because in this method we evaluate the spatio-temporal derivatives of the signal numerically using a finite-differences approach. This computation might result in very high derivatives in these non-smooth locations and interfere with the optimization task. Although CONFIDE was not designed specifically for this task, a straight-forward solution would be to evaluate if the observed signal has non-smooth locations, and to not include the derivatives evaluated in these locations in the optimization task. For example, if $u(t = 0, x = L/2)$ is non-smooth, then the optimization task defined in equation 3 will be performed on the smooth parts separately and summed together. We note that in this test we did not use the above solution since the goal is to test CONFIDE's capabilities in its vanilla form.

For this benchmark, we generated a new test set based on the Burgers' equation experiment, where the initial conditions are not sampled from a single Gaussian process (as described in section B.2), but from two distinct Gaussian processes. The first Gaussian process sampled serves as the left part of the initial condition ($x = 0, .., L/2$) and obeys $u(t = 0, x = 0) = u(t = 0, x = L/2) = 0$, and the second Gaussian process sampled serves as the right part of the initial condition ($x = L/2, ..., L$) and obeys $u(t = 0, x = L/2) = 1, u(t = 0, x = L) = 0$. This formulation creates a discontinuity in the initial conditions at $x = L/2$. We generated 1000 test-signals, all sharing the same characteristics as the Burgers' equations experiment, except all of them have a discontinuity at $u(t = 0, x = L/2)$. In figure 8 we demonstrate two initial conditions of shock-wave signals from the test set. We stress that the train-set is still the original one, since our goal is to test whether CONFIDE is able to handle OOD data, which, in this case, comes in the form of OOD initial conditions.

As shown in figure 9a and in table 3, CONFIDE successfully predicts the given observations and outperforms other baselines, even when it never observed shock-wave signals in the train set. The reason is that CONFIDE evaluates the derivatives at different patches of the observed context $u^c$ at each iteration, and most of them are smooth.

### C.2  OOD COEFFICIENTS

In the second benchmark, we demonstrate how CONFIDE handles the case where the observed signal in the test set is generated from a PDE with coefficients that come from different distribution than the ones in the train set.

For this benchmark, we generated a new test set based on the Burgers' equation experiment, where the coefficient $a$ is sampled from $u \sim U[2, 4]$ instead of $u \sim U[1, 2]$ as in the train set. This modification in the coefficients distribution, results in generated signals that might be significantly different than the ones observed in the train set.

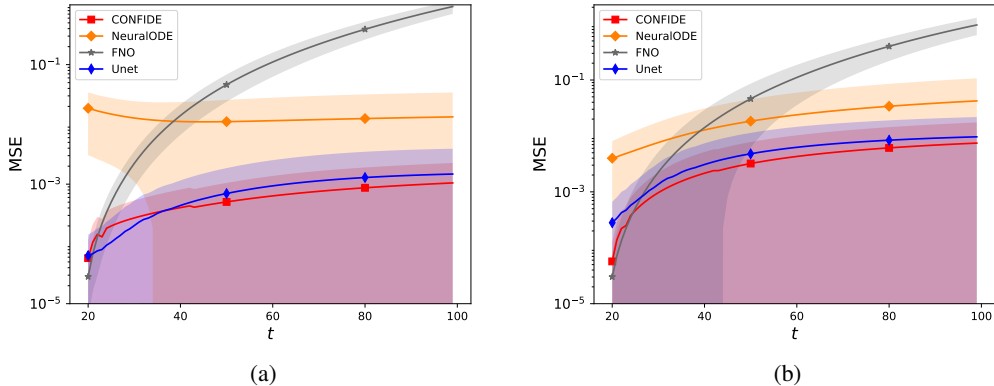

(a)                                            (b)

Figure 9: Prediction error as horizon increases on the two OOD benchmarks for different approaches. In (a) we show prediction results of all approaches on test set that includes a discontinuity in the initial conditions (shock-wave), and in (b) we show prediction results on OOD coefficients (I.e., test set generated from coefficients from different distribution than the ones used in the train set).

As shown in figure 9b and in table 3, CONFIDE successfully predicts the observed signal, and outperforms other baselines. We note that since CONFIDE learns to output coefficients only in the range of the coefficients in the train set, it projects the observed signal to the range of coefficients in the train set so that it best describes the observed signal.

| Algorithm | Prediction MSE shock-wave | Prediction MSE OOD coefficients |
|---|---|---|
| CONFIDE | $0.0010 \pm 0.0012$ | $0.0074 \pm 0.0100$ |
| Neural-ODE | $0.0133 \pm 0.0208$ | $0.0423 \pm 0.0649$ |
| FNO | $0.9367 \pm 0.2322$ | $0.9646 \pm 0.3304$ |
| Unet | $0.0015 \pm 0.0024$ | $0.0096 \pm 0.0121$ |

Table 3: Results summary on the two OOD benchmarks for different approaches.

# D   ABLATION STUDIES

In this section, we provide additional ablation studies to demonstrate how different modifications to the algorithm affect results, both in terms of signal prediction and in coefficient estimation. We start by analyzing the two parameters that characterize the CONFIDE algorithm: train set size and context ratio, using the second order PDE from Section 4.1, and we continue by demonstrating how removing the decoder, or simply removing the initial conditions from the decoder, affect the algorithm's performance.

## D.1   TRAIN SET SIZE

The train set size corresponds to $N$, the number of samples in dataset $U$ of Algorithm 1. Figure 10 presents the decrease in the prediction (panel a) and parameter error (panel b) as we increase the train set size. This attests to the generalization achieved by the CONFIDE architecture: as the train set grows and includes more samples with different values of coefficients, the ability to accurately estimate a new sample's parameters and predict its rollout improves. In this set of experiments, $3,000$ samples are generally enough to achieve a minimal error rate.

## D.2 CONTEXT RATIO

Another hyper-parameter of our system is the context ratio. Figure 11 presents the results of an experiment in which we vary its value as defined in Section 3.3. Simply put, as the context size increases, CONFIDE encodes more information regarding the input signal's dynamics, thus the improvement in signal and parameter value prediction. The error decreases rather quickly, and a context ratio of $0.15 - 0.2$ suffices for reaching a very low error, as is evident from the plots.

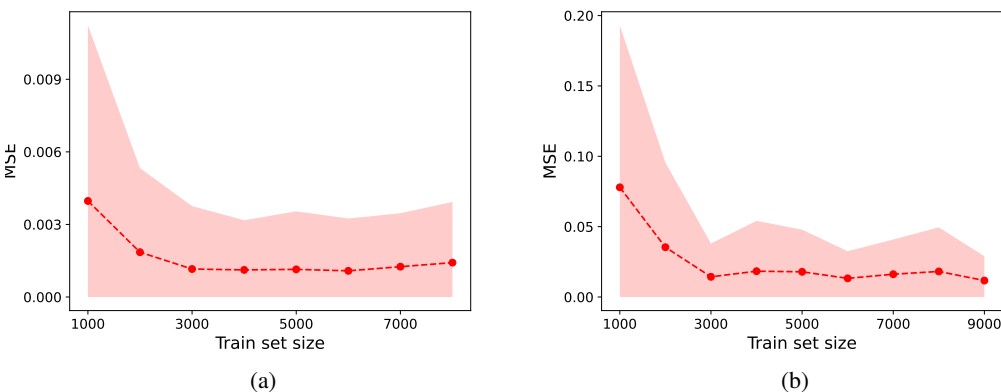

Figure 10: Constant coefficients PDE: **(a)** Prediction error of signal vs. train set size and **(b)** estimation error of parameter values vs. train set size. The error is calculated on a test set of 1000 samples.

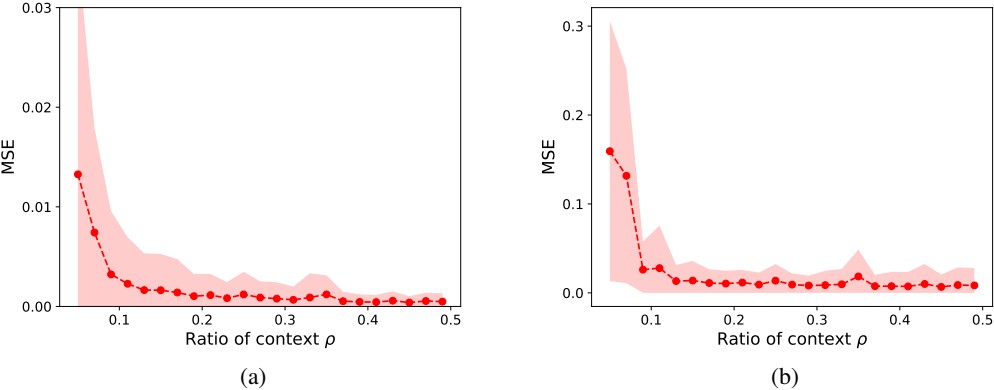

Figure 11: Constant coefficients PDE: **(a)** Prediction error vs. $\rho$ and **(b)** estimation error of parameter value vs. $\rho$. The error is calculated on a test set of 1000 samples.

## D.3 AUTOENCODER

In this section, we demonstrate the effect that adding a decoder network has on CONFIDE. To this end, we evaluate three different scenarios:

- **CONFIDE.** Using a decoder followed by a reconstruction loss, and feeding the initial conditions in addition to the latent vector (demonstrated in the text as initial conditions aware autoencoder)

- **AE-IC.** Similarly, using a decoder followed by a reconstruction loss, but the autoencoder is not initial-conditions aware.

- **No-AE.** The network trains solely on the PDE loss, without the decoder part (i.e., by setting $\alpha = 0$).

Results of the three approaches on the constant PDE dataset are shown in Fig. 12. When comparing a setup with no decoder part (i.e., No-AE) with a setup that has a decoder, but does not use the initial conditions as a decoder input (i.e., AE/IC), we observe that merely adding a decoder network might have a negative effect on the results, especially when analyzing the parameter estimation results. One reason for this may be that the neural network needs to compress the observed signal in a way that should both solve the PDE and reconstruct the signal. This modification of latent space has a negative effect in this case. When also adding the initial conditions as an input to the decoder (i.e., the standard CONFIDE), we observe significant MSE improvement in both signal prediction and parameter estimation ($\sim$35% improvement in both). This result suggests that adding the initial conditions aware autoencoder enables the networks to learn a good representation of the dynamics of the observed signal in its latent space.

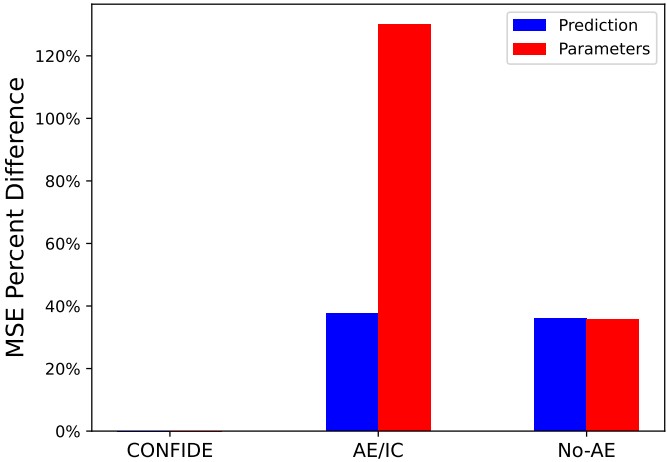

Figure 12: Ablation study on the constant coefficients PDE dataset. The Y axis shows the percentage difference between the different approaches and the standard CONFIDE one (thus it scores 0%). We demonstrate the effects on both signal prediction (blue), and parameter estimation (red).

## E    DIFFERENTIATION VS INTEGRATION APPROACHES

A key part in the CONFIDE algorithm is its use of numerical evaluation of spatio-temporal derivatives through the finite-differences approach. As shown throughout this paper, using finite differences combined with the a-priori known mechanistic form of the PDE provides both coefficients estimation and reliable predictions. However, there's another important advantage that a finite-differences approach has on other integral approaches (such as the adjoint method used in Neural-ODE (Chen et al., 2018) and other relevant works), the amount of time necessary for training the model.

To demonstrate this effect we create a toy example based on an ODE of a single pendulum without friction:

$$\ddot{\theta} = -\frac{g}{l}\sin(\theta),$$

where $g$ is the gravitational parameter, $\theta$ is the angle of the pendulum, and $l$ is the length of the pendulum. We created a train set of 10,000 signals that obey the pendulum ODE, with the length parameter $l$ sampled for each signal $l \sim U[1, 2]$, the initial conditions of $\theta(t = 0)$ sampled from $\theta \sim U[-0.4, 0.4]$ and the initial conditions of the angular velocity $\dot{\theta} = \omega \sim U[-0.1, 0.1]$.

The two algorithms that are tested are CONFIDE and a physics-informed (PINN) version of Neural-ODE. We used the PINN version of Neural-ODE so that both algorithms are presented with the same

information and have the same goals: to estimate the length of the pendulum on a given signal $l$, and provide the observed signal future prediction $\theta(t > T)$, where the observation is up to time $T$.

Both algorithms use a single neural-network that consumes an observed signal, and generates an estimation $\hat{l}$. The next step of both algorithms is to enforce that $\hat{l}$ can generate the observed signal. CONFIDE uses a differentiation approach by numerically evaluating the time derivative of the observed signal, and comparing this derivative to the function of the ODE. I.e., minimizing the objective:

$$\min \left\| \frac{d^2\theta}{dt^2} - \frac{g}{\hat{l}} \sin(\theta) \right\|,$$

where $d^2\theta/dt^2$ is evaluated through finite differences. At inference time, the observation is consumed by the neural-network, which outputs $\hat{l}$. We then feed $\hat{l}$ and the initial conditions to an ODE solver to obtain the prediction. Neural-ODE uses the initial value ($\theta(t = 0)$) and $\hat{l}$ to generate the signal $\hat{\theta}(t = 0, .., T)$, and compare the generated signal to the observed one using an MSE loss.

We trained both algorithms in exactly the same setting, over 5 epochs, and observed that CONFIDE train significantly faster. Specifically, CONFIDE's training time took 3.6 seconds, and neural-ODE's training time took 159.2 seconds. I.e., **CONFIDE trained** $\sim 44$ **times faster than Neural-ODE**.

In figure 13 and in table 4, we see that not only did CONFIDE train much faster, it also was able to provide very reliable results, both in terms of prediction and in terms of parameter estimation.

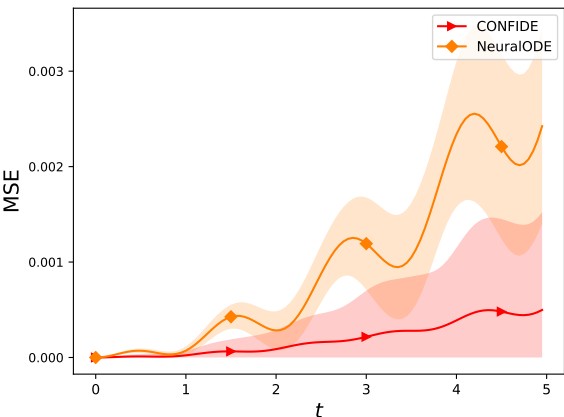

Figure 13: Pendulum ODE: prediction error as horizon increases, for different approaches.

| Algorithm | Prediction MSE | Coefficient MSE | Training time |
|---|---|---|---|
| CONFIDE | $0.0002 \pm 0.0003$ | $0.1598 \pm 0.1891$ | 3.6s |
| Neural-ODE | $0.0010 \pm 0.0003$ | $0.1702 \pm 0.2021$ | $159.2s$ |

Table 4: Results summary on the pendulum ODE in terms of future time prediction error, pendulum length estimation error and training time between two different approaches: a differentiation approach (CONFIDE) and an integral approach (Neural-ODE)

