# OpenReview forum: "CONFIDE: CONtextual FInite DifferencE modelling of PDEs"
_ICLR.cc/2024/Conference — Submitted to ICLR 2024_

### Official Review · Reviewer_EEcP · 2023-10-30

**Soundness:** 3 good
**Presentation:** 3 good
**Contribution:** 2 fair
**Rating:** 5
**Confidence:** 4

**Summary:**

This paper solves a family of PDEs given their general functional form. The model first derives the PDE coefficients from the encoded input data and then solves the PDE using some off-the-shelf numerical PDE solvers. This method is tested on several families of PDEs and shows some generalization ability.

**Strengths:**

1) Clarity: Detailed description of the proposed method, the experiment protocols datasets involved, and the baselines.
2) Originality: Estimate PDE coefficients by inference rather than optimization.

**Weaknesses:**

1) Novelty: Both the task definition and network architecture are not novel. The task of estimating PDE coefficients from observation data is previously known as the PDE inverse problem. And the AE architecture and finite difference scheme are not new.

2) Significance: the main contribution is claimed as 'Proposing a DL encoding scheme ... enabling generalization for prediction of unseen samples based on minimal input'. However such generalization was only tested on in-distribution settings. Also, the expensive time cost of incorporating a finite-difference solver is not addressed or discussed.

3) Solidness: the authors claimed in the abstract that 'We include results of extensive experimentation, comparing our method to SOTA approaches...'. however, the PDEs in experiments are relatively simple, and the baselines are not strong, compared with [1].

[1]Wu, Haixu, et al. "Solving High-Dimensional PDEs with Latent Spectral Models." (2023).

**Questions:**

1) For params estimation, is the generalization or time cost of CONFINDE better than PINN and the classical adjoint method? For PDE solving, is the time cost of CONFINDE smaller than pure neural models such as FNO?

2) The experiments are all about smooth solutions (Fig3 and Fig5). But in more practical settings, the PDE solution includes a discontinuity, also known as a 'shock wave' in Burgers and N-S. In such cases, the finite difference is more likely to fail or require more computations than in the smooth solution. Can you add some experiments on the shock wave data?

3) Can you test the 'zero-shot' in the OOD setting, or at least in-distribution on a larger domain? For example,  change the Burgers parameter $a \sim U[0,1]$ to $U[0,10]$.

4) In the ablation study,  network size (number of learnable parameters) varies in different settings, which may affect the validity of the performance comparison.

---

> ### Author Response · Authors · 2023-11-19
>
> Thank you for the helpful feedback. We are pleased to report that we added the suggested experiments to the new version of the paper.
>
> **Regarding weaknesses:**
> 1. While we agree that estimating PDE coefficients from observations is not a new task, the task we aim to solve is quite different. In our scenario, each signal in the datasets (both train and test) are generated from different dynamics. This means that there are dynamics in the test set never seen before. Having different coefficients per signal makes our task much more challenging than other related works, which mainly focus on the scenario where the entire dataset is created from the same dynamics but with different initial conditions.
> The novelty in our approach is that it is capable of creating reliable predictions even in this scenario, while also providing a specific estimation of the PDE coefficients for the given signal. This estimation holds high value in many scenarios as also discussed in many other papers in the area (for example papers [1] and [2].
> 2. We agree that using a PDE-solver has an expensive time cost. However, the main goal of our paper is not to create a fast replacement to a solver (such as in neural-operator related works). Our algorithm’s main goal is to utilize given mechanistic knowledge in order to estimate the coefficients for each signal. We then show at inference time how these coefficients could be used to provide reliable predictions.
> 3. As we show throughout our experiments section, a relatively simple PDE is already a hard task when considering every signal in the dataset comes from a different context. This task gets even harder when the test set signals are generated from coefficients that were never seen in the train set.
> Regarding the benchmarks suggested in the paper you provided, all datasets are generated from using the same coefficients. For example, in the Navier-Stokes experiment, the viscosity coefficient is constant (10^-5), and in the Darcy-Flow experiment, the external force equals 1 for all signals.
>
> [1] Linial et. al. "Generative ODE modeling with known unknowns"
> [2] Yang et. al. "Learning Physics Constrained Dynamics Using Autoencoders"
>
> **Regarding questions:**
>
> 1. For params estimation, our method provides a very fast alternative to the “integral” approach used in the adjoint method. Adding a PINN loss does not improve the time cost of this method.
> To demonstrate this, we added a new demonstration to the appendix (appendix E), where we demonstrate how CONFIDE works in comparison to an adjoint+PINN method.
> In this experiment, we evaluate these methods on a pendulum equation (which is formulated as an ODE), and demonstrate that CONFIDE’s results are not only better in terms of prediction and coefficient estimation, but also trains ~44 times faster.
> 2. As requested, we added a new shockwave experiment based on the Burgers’ equation in appendix C1.
> For this experiment, we created a new test set of signals that have a discontinuity at t=0, and compared the algorithms on this data. We observed that CONFIDE still outperformed other baselines (as can be seen in table 3). We believe that the reason is that CONFIDE evaluates the derivatives at different patches of the observed context $u^c$ at each iteration, and most of them are smooth.
> Note that in this case, CONFIDE could be modified to handle discontinuity, by evaluating the spatial and temporal derivatives only on the parts in the signal that are smooth. For instance, if we observe that there is a discontinuity in the initial conditions of the signal, at x=x_0, then we can evaluate finite differences for x<x0, and x>x0 separately and enforce the PDE in these regions. For the sake of evaluation we did not add this modification to our algorithm in this
> 3. As requested, we added a new OOD experiment based on the Burgers’ equation in appendix C2.
> For this experiment, we created a new test set, where in all signals the coefficient a is sampled from U[2,4] instead of U[1,2] as in the train set.
> Note that we did not re-train any of the algorithms, but used the ones trained on the a~U[1,2].
> As can be seen in the results, CONFIDE still outperformed other baselines although it never observed coefficients as these.
> CONFIDE projects the observed signal to the range of coefficients in the train set, and finds the ones that best describes the observed signal.
> 4. In the ablation study where we change the size of the context (the amount of time steps in each signal in the train set), the only change of size is in the first layer. The network size does change but not drastically:
> rho=0.05: #params = 4.7M,
> rho=0.2: #params = 5M,
> rho=0.5: #params = 5.6M.
> As can be seen in figure 11, CONFIDE’s results are almost not impaired even when the context size is very small, and the network size is a bit smaller as a result.

---

> > ### Comment · Reviewer_EEcP · 2023-11-20
> >
> > Thank you for the detailed reply, extended experiments, and the updated paper. I raised my score to 5, but I still have the following unaddressed questions:
> >
> > 1. w.r.t. Q1 and appendix E.  Sorry for my ambiguous expression "...CONFINDE better than PINN and the classical adjoint method". I mean both PINN and adjoint methods can solve inverse problems. PINN paper [1] Figure 4 shows that it can recover NS equation coefficients. Your newly added experiments show that CONFIDE is faster than the adjoint method or physics loss+adjoint method, but not the PINN.
> >
> > 2. w.r.t. Q2 and appendix C.1.  You created a non-propagating artificial discontinuity in the initial condition and handled them "by evaluating derivatives only on the smooth parts at t=0". However, the usual shock wave is generated and propagated by the fluid itself ([wiki link](https://en.wikipedia.org/wiki/Shock_wave)). How to predict/solve discontinuities generated by the fluid itself at some future t>0? And how to deal with propagating discontinuities? I still think the finite difference, at least your forward-time central-space scheme, is not a suitable solver for practical PDE problems, which limits the significance of your work.
> >
> >
> > Ref:
> > 1. Raissi, M., Perdikaris, P. & Karniadakis, G. E. Physics-informed neural networks: A deep learning framework for solving forward and inverse problems involving nonlinear partial differential equations. Journal of Computational Physics 378, 686–707 (2019).

---

> > > ### Author Response · Authors · 2023-11-21
> > >
> > > Thank you for the clarification.
> > >
> > > 1. As requested, we added PINN for this experiment. It is worth mentioning that PINN (as displayed by Raissi et. al.) is not designed to solve the task we are dealing with. First, the coefficients in the discovery section are all constants, and second their model is trained to fit  only a single signal (part of it is used as training and the rest is used for test).
> > > To overcome this difference, we iterated through the test set, and trained an entirely new model on every observed signal. This resulted in the estimated parameters for each signal.
> > > This training procedure is extremely slow (took ~517.2 sec, which is even more than NeuralODE), since there is no knowledge sharing between signals in the train set, and the model needs to be fitted to the test signal. Moreover, after carefully implementing several versions, the PINN algorithm did not provide results that are close to the other algorithms. This happened because the PINN algorithm was able to find a different set of parameters that balanced between the physics loss and reconstruction loss. Results are: prediction error 0.1564 +- 0.1570, parameters error 1.6471 +- 0.7752.
> > >
> > > 2. We are sorry for the confusion. In this experiment we did not handle the task by evaluating derivatives only on the smooth parts at t=0, but only *suggested* that this could be a valid solution. Our suggestion was that every point in the observation (for any location and time) that has a discontinuity should not participate in the loss function, as long as we know about its existence in advance.
> > > In our discontinuity experiment, we let CONFIDE handle the discontinuity without any modification to the original algorithm. CONFIDE handled the problem successfully, as evident in the results.

---

> > > > ### Comment · Reviewer_EEcP · 2023-11-22
> > > >
> > > > Thank you for the additional experiments and efforts. I decided to keep my score.
> > > >
> > > > 1. It is true that PINN (and other neural solvers) have longer offline training time, but they generally have lower online inference time, compared with classical numerical schemes. In real applications, the inference speed is usually more critical. As for the drawback that the model "is trained to fit only a single signal" and "has unbalanced two losses", there are also some (non-SOTA) remedies [1,2]
> > > >
> > > > 2. To my knowledge, the vanilla finite difference is not a reliable scheme for a simple 1D Euler's equation (the shock tube problem, the simplest benchmark of CFD solver), see the demonstration in Section 1.4 in [3]. Can CONFIDE handle such a problem?
> > > >
> > > > Ref:
> > > > 1. Huang, Xiang, et al. "Meta-auto-decoder for solving parametric partial differential equations." Advances in Neural Information Processing Systems 35 (2022): 23426-23438.
> > > > 2. Wang, Sifan, Yujun Teng, and Paris Perdikaris. "Understanding and mitigating gradient flow pathologies in physics-informed neural networks." SIAM Journal on Scientific Computing 43.5 (2021): A3055-A3081.
> > > > 2. LeVeque, Randall J., and Randall J. Leveque. Numerical methods for conservation laws. Vol. 214. Basel: Birkhäuser, 1992.

---

> > > > > ### Author Response · Authors · 2023-11-22
> > > > >
> > > > > Regarding (2), first, in a general sense, CONFIDE should be able to handle a shockwave problem to some extent.
> > > > > If the observed signal has some regions (either time or space) with dynamics that are effected by the coefficients, it should be able to find them.
> > > > >
> > > > > The Euler's equation example, is indeed a hard task since almost all the derivatives are zero, or not continuous.
> > > > > While Euler’s equation is indeed an interesting benchmark for PDE solvers, the problem we present is generally different. We do not focus on signals that are specifically hard to solve, but focus on the scenario where the observed signals are generated by different coefficients, and we want to find them and generate future predictions.
> > > > > We hope that our problem formulation could inspire other researchers in the field to create new algorithms for these tasks, and to integrate advanced finite-differences schemes into CONFIDE.

---

### Official Review · Reviewer_RwBU · 2023-10-31

**Soundness:** 2 fair
**Presentation:** 2 fair
**Contribution:** 2 fair
**Rating:** 5
**Confidence:** 4

**Summary:**

The authors propose a method to infer PDEs from data generated by unseen dynamics. The authors claim to tackle the lack of explainability of DL models for PDEs in comparison to numerical methods and their inability to extrapolate well to unseen data. To do so, the authors propose a hybrid approach CONFIDE. The method learns the unknown PDE parameters and then with the parameters solves the PDE forward in time.

**Strengths:**

- Nice overview, background and motivation for PDEs in science and engineering applications.
- Nice overview of numerical methods and mentioning RANs scheme.
- The authors identify real limitations of current DL models for PDEs: the lack of explainability and inability to extrapolate to unseen data. Both are very serious limitations of these methods in practice for real-world science problems.
- Nice hybrid approach and good to bring methods from numerical analysis, e.g., finite differences to black-box DL models.
- The figures and plots are well-generated and easy to visualize.
- It is good that the authors compare to FNO and U-Nets but I would also like to see comparisons to MeshGraphNets (Pfaff et. al, "Learning Mesh-Based Simulation with Graph Networks", ICLR 2021) and PINNs.
- It is nice that the authors identified the issue common in many DL methods for PDEs that the error accumulates over time, such as in the observed results with FNO and U-Net.

**Weaknesses:**

- Abstract is a bit unclear
- Last sentence of first paragraph in introduction is a bit vague.
- The second paragraph in the introduction is also vague. Learning a PDE directly from data is a very different task than assuming it has a specific structure, such as the heat equation referenced here and just learning its coefficients. The description of the task is really the latter simpler task of learning the coefficients and is buried inside the introduction and is difficult to follow.
- More examples are needed on why modeling using PDEs is limited. The (in)compressible Navier Stokes equations has been state-of-art in ocean dynamics and aerodynamics, respectively. I can see the motivation for not knowing the PDE parameter value exactly.
- Numerical methods seem to be discounted and credited only for low-complexity problems but they are still state-of-the-art and have been made efficient with the last 40-50 years of high performance computing (HPC) research and numerical analysis. Good mention and analogy to RANs models though.
- Too much description of the method in the introduction section that should be saved for the method section.
- This two stage approach is reminiscent of CROM (https://arxiv.org/pdf/2206.02607.pdf), ICLR 2023, which uses neural representations to represent the spatial part of the PDE and then solves the resulting semi-discrete ODE forward in time using time stepping method.
- The choice of finite difference methods over finite volume (suited for hyperbolic conservation laws) and finite element methods for unstructured meshes.
- The background on current state-of-the-art SciML methods is extremely limited in the second paragraph of the related work section. In particular, Brandstetter et al., 2022 and Li et al., 2020 are very different methods, where the former is message-passing GNN based and the later is a Neural Operator approach. An advantage of Neural Operators, which is not discussed here is that they can learn the mapping from PDE parameters to solutions (Subramanian et. al, "Towards Foundation Models for Scientific Machine Learning: Characterizing Scaling and Transfer Behavior", NeurIPS 2023 and Negiar et. al, "Learning differentiable solvers for systems with hard constraints", ICLR 2023).
- Reference to MeshGraphNets (Pfaff et. al, ICLR 2021) is missing which is another state-of-the-art class of GNN models for PDEs.
- PINNs (Raissi et. al) can be used to solve inverse problems.
- The authors should not be discussing details of their method in related work before describing their method in detail.
- Assuming the form of the PDE is still a strong assumption.
- Since the authors are comparing to a variant of NeuralODE, they should compare to DINO ("Continuous PDE Dynamics Forecasting with Implicit Neural Representations", ICLR 2023), which is designed for PDEs instead of ODEs for NeuralODEs and uses neural representation for the spatial discretization and then NeuralODEs to advance the continuous ODE forward in time.
- The authors note that they cannot use PDEBench in the experiments since it is generated from the sample context. For other benchmarking examples see the GPME benchmarking framework in Hansen et. al, "Learning Physical Models that can respect conservation laws", ICML 2023 and various challenging PDEs including Navier-Stokes in Saad et. al, "Guiding continuous operator learning through Physics-based boundary conditions", ICLR 2023.
- Large limitation to only test on (quasi)-linear and not concern challenging nonlinear hyperbolic PDEs is where numerical methods actually have challenges, e.g., hyperbolic conservation laws (nonlinear PDEs) and Stefan in Hansen et. al
- The x vs t heat maps are more difficult to visualize in Figure 3 than the solution over space plotted at particular times as done in numerical methods. Also these plots are too qualitative to visual the performance improvement and plots of the errors would be better. Also it may be better to move these solution profiles to an appendix. Similarly for Figure 5.
- Burgers' should be labeled properly as quasi-linear
- The authors mention their motivation is for more explainable solution to PDEs and handle extrapolation. I don't see how either of these objectives are met by the method or shown in the experiments. The abstract should be rewritten to clarify this.

Minor
- Put RANs in parenthesis
- Takamoto reference for PDEBench should be in parenthesis and similar for other citations

**Questions:**

1. How does learning the PDE parameters couple with and affect the accuracy of the numerical solver?
2. The numerical solver is a critical component and several solvers are better for different PDEs. Why was PyPDE chosen and which numerical schemes is it using?
3. Why are finite difference methods used over finite volume and finite element methods? The choice of numerical method may vary for the different class of PDEs. Also, I don't understand why using finite differences eliminates the need to explicitly state the boundary conditions.
4. Why is an autoencoder architecture selected? Were there ablation studies to motivate this choice?
5. Based on the notation in Eqn. (1), does the method only work on linear PDEs? For instance, I see nonlinear Burgers $u_t + uu_x = 0 \iff u_t + \frac{1}{2}\frac{\partial u^2}{\partial x} = 0$ in the experiments but I don't see how this equation fits in the form of Eq (1). I later found in Section 4.2 that it is only quasi-linear Burgers that is concerned so I think the p in Eqn. (1) could not be constant and need to depend on $u$ even in this case.
6. Why is FNO and U-Net beating the proposed method CONFIDE in Figure 2a at early time steps?
7. Why is FNO performing so poorly on the Fitz-Hugh-Nagumo equations? This seems odd.
8. Since CONFIDE is not an operator, is it resolution invariant like Neural Operator methods are?
9. Is the extrapolation task tested in the experiments?
10. How is CONFIDE an "explainable" method?

---

> ### Author Response · Authors · 2023-11-19
>
> Thank you for the attentive and informative feedback.
>
> **Regarding Weaknesses:**
>
> As per requested, we have added some modifications to the text (marked with red), and changed figure 3 to display the solution at particular time points.
>
> **Regarding questions:**
>
> 1. The process of learning the PDE parameters does not depend on the numerical solver at all. During the training phase, our model consumes the observed signal and outputs an estimation for the coefficients. These coefficients are then optimized to best fit the PDE alongside the derivatives which are numerically computed. It is only in the inference phase where we might use a numerical solver in order to provide predictions to a given signal.
> 2. The main goal of our paper is not to create a new efficient PDE solver (such as in neural-operator related works), but to utilize given mechanistic knowledge in order to estimate the coefficients of the PDE (which are different for each signal). We then demonstrate how one can use the estimated coefficients to create reliable predictions, even using a rather simple solver such as PyPDE (specifically the explicit solver).
> Our training scheme does not use any solver, but directly infers the PDE coefficients.
> 3. We use finite-differences method only to obtain numerical derivatives which can be substituted into the PDE. Using other methods, as you mentioned, is completely valid as long as the derivatives can be calculated. The choice of finite-differences in our case, was enough to result in reliable and fast estimation of the coefficients. However, a user can change this easily and plug-in any choice of derivatives calculations.
> Regarding boundary-conditions: when using finite differences to compute the derivatives, we do not use the boundary condition (we only need them when attempting to solve the PDE). Therefore, estimating the coefficients does not depend on boundary-conditions. However, during inference time, when a user wants to use the estimated coefficients to solve the PDE, one would need to determine what boundary conditions are needed.
> 4. We address this question in ablation appendix C.3 (figure 10) (which in the new version became D3, and figure 12), where we compare between an architecture without a decoder part, one with a decoder part and one with a decoder part which is also “initial-conditions aware”.
> 5. We are sorry for the confusion. The coefficients in equation (1) are not necessarily constants, but could be functions. In the Burgers’ experiment, the coefficient b is a function of u, and in the FN2D experiment, the coefficients which are the local reaction terms depend on both solution signals u and v.
> 6. Our method learns a mapping between the observed signal and the coefficients. Estimating the coefficients correctly results in good long-horizon prediction as shown in the figure (and as usually desired). FNO and UNET train to predict one-step into the future, and then auto-regressively iterate to produce the next predictions. This is why these methods tend to do well in the very near future predictions, while CONFIDE tends to be more robust and reliable.
> 7. As can be seen in Figure 6, FNO behaves very similarly to UNET through most of the prediction time until at some point it diverges from the true solution. The prediction figures we added should help to understand how each algorithm scored at each time point.
> 8. CONFIDE could be used as long as numerical derivatives can be computed (e.g., by finite-differences). This notion does not change when resolution changes, therefore there is no limit to using CONFIDE on different resolutions.
> 9. We stress that each signal in our test set is generated not only from different initial conditions, but also from coefficients not seen in the train set. As can be seen in our experiments section, this task alone is already a hard task, as modern baselines struggle to generate reliable predictions.
> 10. We note that the term “explainable” here is not the same as standard “explainable-ML” methods.
> Our meaning was that since CONFIDE provides coefficient-estimation to a given signal, it provides the user some knowledge (or explanation) regarding the observed signal. For example, In the electric vehicle batteries case ( last paragraph of the introduction), the coefficients hold valuable information regarding the battery’s capacity, how old it is, how fast it could discharge and so on. In healthcare for instance, there has been a great effort in designing a differential system of the cardio-vascular system, and inferring the parameter of cardiac contractility, or the parameter of arterial resistance, holds great value for clinicians as they decide on their treatments.

---

> ### Author Response · Authors · 2023-11-21
>
> In addition to the last comment, we have added the DINO baseline as you suggested.

---

### Official Review · Reviewer_VkbV · 2023-11-03

**Soundness:** 3 good
**Presentation:** 3 good
**Contribution:** 3 good
**Rating:** 6
**Confidence:** 2

**Summary:**

This paper proposes a hybrid model for PDEs. The model combines an autoencoder with prediction of the (dynamically varying) coefficients of the underlying PDE. The method is benchmarked on signals generated with constant and non-constant coefficient PDEs.

I am not fully familiar with this area, so my assessment is primarily based on reading the manuscript itself.

**Strengths:**

- this paper appears to combine the "best of both worlds" in PDE modelling. By combining autoencoding with coefficient prediction, it uses more PDE information than purely data-driven approaches but does not have the limitation of assuming constant PDE coefficients
- the method is clearly explained
- the results appear to  be competitive or better than existing methods such as U-Net

**Weaknesses:**

- the method is limited to PDEs of the form (1)
- no comparison with baselines such as Brandstetter et al., 2022 on prediction accuracy

**Questions:**

-

---

> ### Author Response · Authors · 2023-11-19
>
> Regarding weaknesses:
> 1. The CONFIDE algorithm is not necessarily limited to the form of equation (1).
> For comparison and demonstration purposes we assumed the form in (1), but in the general case CONFIDE could handle any PDE form, as long as numerical derivatives could be computed through finite-differences. For example, CONFIDE has the ability to evaluate the coefficient in the wave equation (d^2u/dt^2 = c^2 * d^2u/dx^2)  by evaluating the second derivatives of time and space on the observed signal u(t=0..T, x=0…L).
> 2. For comparison we chose the SOTA algorithms currently existing (FNO and UNET), and added a baseline that demonstrates an integral approach vs differential approach.

---

### Author Response · Authors · 2023-11-19

We would like to thank all of the reviewers for their thoughtful and thorough feedback. After carefully considering the feedback from all of the reviewers, we are pleased to report that we have several experiments as requested:
1. Demonstration on shockwave data.
2. Evaluation on OOD test data, where the coefficients that were used to generate the data came from a different distribution than the coefficients used to train the data.
3. Added an additional experiment that shows the time cost differences between a differential approach (CONFIDE) vs an integral approach (such as Neural-ODE with the adjoint method).

We also modified the text in light of the reviewers suggestions (modifications colored in red).

---

### Author Response · Authors · 2023-11-21

In addition to the last revision, we added the DINO [1] algorithm, an additional baseline requested by the reviewers.
We ran DINO on two out of the three experimental setups. Running it on the FN2D setup requires additional coding effort, which we are unable to achieve during the rebuttal period. We will therefore add it to the camera ready version of the paper.
As can be seen in the results, DINO scores roughly as well as our version of NeuralODE, and both are outperformed by CONFIDE.

[1] Yin et. al. "Continuous PDE dynamics forecasting with implicit neural representations" (2023)

---

### Meta-Review · Area_Chair_xFXh · 2023-12-04

**Metareview:**

This technical paper proposes a new method to learn an explicit PDE from data arising from unobserved dynamics. It is a well written paper but ultimately the reviews are borderline and it is not clear how big a step this takes compared to existing work to solve these inverse problems. The reviewers appreciated the detailed rebuttal but there is simply not as much excitement as other strong submissions this year.

**Justification For Why Not Higher Score:**

Borderline reviews

**Justification For Why Not Lower Score:**

na

---

### Decision · Program_Chairs · 2024-01-16

Reject